# Compound Context-Aware Bayesian Inference Scheme for Smart IoT Environment

**DOI:** 10.3390/s22083022

**Published:** 2022-04-14

**Authors:** Ihsan Ullah, Ju-Bong Kim, Youn-Hee Han

**Affiliations:** 1Advanced Technology Research Center, Department of Computer Science and Engineering, Korea University of Technology and Education, Cheonan 330-708, Korea; ihsan@koreatech.ac.kr; 2Future Convergence Engineering, Department of Computer Science and Engineering, Korea University of Technology and Education, Cheonan 330-708, Korea; rlawnqhd@koreatech.ac.kr

**Keywords:** smart IoT environment, sensor data fusion, context awareness and sharing, Bayesian networks, Kalman filter, smart cities

## Abstract

The objective of smart cities is to improve the quality of life for citizens by using Information and Communication Technology (ICT). The smart IoT environment consists of multiple sensor devices that continuously produce a large amount of data. In the IoT system, accurate inference from multi-sensor data is imperative to make a correct decision. Sensor data are often imprecise, resulting in low-quality inference results and wrong decisions. Correspondingly, single-context data are insufficient for making an accurate decision. In this paper, a novel compound context-aware scheme is proposed based on Bayesian inference to achieve accurate fusion and inference from the sensory data. In the proposed scheme, multi-sensor data are fused based on the relation and contexts of sensor data whether they are dependent or not on each other. Extensive computer simulations show that the proposed technique significantly improves the inference accuracy when it is compared to the other two representative Bayesian inference techniques.

## 1. Introduction

The rapidly growing world population is becoming a relevant issue to solve the problems of efficiency and quality of life for people [1,2]. Smart cities use the latest technologies to improve citizens’ economic growth and lifestyles. The key features of smart cities include: smart governance, smart monitoring, smart citizens, smart services, smart economy, smart technology, smart mobility, smart living, smart environments, and smart parking [3,4,5]. IoT systems and Information and Communication Technology (ICT) play a vital role to increase intelligence for making better decisions based on the sensor data environments [6]. It is based on the paradigm of sensing, reasoning, inferencing, and acting by exploiting the sensory data [7,8,9].

The smart city comprises different types of sensor devices which often produce diverse data. In such a situation, it is necessary to fuse the heterogeneous data obtained from various sensors deployed in the target place. However, it is challenging to accurately infer and make a correct decision based on the multi-sensor data. Furthermore, data from a single source (sensor) are usually insufficient for making the correct decision. Additionally, noisy data may result in wrong inferences about the environment [10]. Thus, noise in the sensor data needs to be filtered out before the data are forwarded to the inference system. We adopted the Kalman filter (KF) which is the most commonly used technique to reduce the noise and uncertainty in data [11].

Context awareness allows accurate inference by properly interpreting the context information extracted from the sensor data in an integrated fashion, either passively or actively [12,13]. The IoT system of smart cities also needs compound context awareness to achieve accurate decisions since only one context may lead to a wrong inference. For example, high body temperature in an ordinary situation indicates illness, but it is normal during strenuous exercise. Hence, two contexts, temperature and location, may be needed to distinguish illness from exercise. The techniques used for multi-sensor data fusion and inference are mainly categorized as artificial-intelligence-based, evidence-based, and probability-based [14,15,16]. Bayesian inference (BI) is a probability-based fusion technique that obtains the correlation between the multi-source data. In the literature, various data fusion and inferencing techniques have been proposed [17,18,19,20,21]; however, they are mostly based on a single context. The appropriate manipulation of compound contexts for accurate inference is still a big challenge, which is the motivation of our research and proposed scheme.

In this paper, a compound context-aware Bayesian inference (CCBI) scheme is presented which allows accurate inference and decision making based on multiple-context data sharing of the smart cities environment. Multi-sensor data are operated in two phases; in the first phase, KF is applied to filter each sensor datum and reduce the error. In the second phase, BI is applied on filtered data to exploit the statistical correlation between the sensor data based on the multiple contexts and fuses them. Altogether, in the CCBI, the heterogeneous data from multiple sensor devices are fused based on context sharing and relation to perform an accurate decision on them. Computer simulation reveals that the CCBI shows considerably better performance than the other two schemes, Bayesian Data Fusion [22] and Bayesian Network Data Fusion (BNDF) [23], in terms of inference accuracy. The following are the main contributions of the proposed scheme:While various multi-sensor data fusion schemes with BI have been proposed, they are mostly based on a single context. We proposed a generic approach for improving reasoning and inference accuracy by sharing and utilizing multiple compound contexts.Since the events in the real situation might be correlated with each other, the inference operation is further specified in two modes to best match the given condition between the contexts of sensor data: (i) Bayesian inference with dependent contexts and (ii) Bayesian inference with independent contexts.A novel belief function of the BI system is developed which effectively represents the conditional dependency between a specific state and contextual information. The proposed modeling approach is general so that it can be adopted for any inference problem handling heterogeneous data.

The organization of the paper is: the work related to multi-sensor data fusion for WSN and smart cities is summarized in Section 2. In Section 3, the proposed CCBI scheme is discussed. Simulation setup and results analysis are explained in Section 4, and the conclusion is discussed in Section 5.

## 2. Related Work

The issue of sensor data fusion in smart IoT environments has been recognized by several researchers. In [24], a platform named iSapiens was proposed to apply the edge computing model concerning a distributed network in an urban environment. It was an IoT-based application platform to develop agents for receiving data from the smart environment and situation. In [25], a deep learning-based scheme was presented where a convolution neural network (CNN) was incorporated with long short-term memory (LSTM) architectures to be used for traffic flow predictions in smart cities. Spatial data were classified with a CNN, though temporal data were classified with LSTM. In [26], an adaptive distributed Bayesian was proposed to detect outliers in the sensor data.

In [27], the authors presented a scheme for data fusion using reinforcement learning to improve fusion accuracy. Ref. [28] presented a deep-learning-based method for fused multi-source heterogeneous data. Correspondingly, several issues with multi-source heterogeneous data fusion were discussed in this paper. A Bayesian-based model [29] was proposed for data fusion that measures temperature, fuses data from smart buildings, extracts knowledge with some sensor measurements, and then predicts spatial temperature distribution for further estimation. The Bayesian information and knowledge fusion model is presented in [30], to maximize the posterior probability. In Equation (1) [30], the Bayesian approach for information fusion is formulated.
(1)$$p(\Theta |D_{1},D_{2},|M_{1},M_{2})=\frac{p\left(D_{1}|\Theta ,M_{1}\right)p\left(D_{2}|\Theta ,M_{2}\right)\;P\left\{p\left(\Theta |M_{1}\right)p\left(\Theta |M_{2}\right)\right\}}{p\left(D_{1},D_{2}|M_{1},M_{2}\right)}$$

A fuzzy-neural-network-based technique was presented in [31] to extract the important features and knowledge from the data. Ref. [32] proposes three filtering approaches to fuse sensor data: Pre-Filtering, Post-Filtering, and Pre-Post-Filtering. It filtered and combined sensor data using a modified Bayesian fusion algorithm to deal with ambiguity and contradiction problems in the data. Dempster–Shafer theory is the generalization of the Bayesian theory which is used to fuse and transform conflicting information into a decision-making result [33,34].

## 3. The Proposed Scheme

In this section, we present the proposed CCBI scheme for heterogeneous data fusion and make accurate decisions on them. It increases the inference accuracy based on compound context awareness and sharing.

### 3.1. Design Goal

The design purpose of the proposed scheme is to efficiently fuse the heterogeneous sensor data for accurate inference. The data generated in WSN are usually massive and noisy. Hence, a method to efficiently filter the errors in the sensor data is needed before the fusion. With centralized filterings, such as a base station (BS), the network tends to become congested. So, we consider that each sensor node can accommodate the KF operation for filtering the sensing data to reduce error and noise before transmitting it to the fusion node. The BI operation is deployed at the central node to fuse heterogeneous sensor data. The structure of the CCBI scheme is shown in Figure 1.

### 3.2. Operation

#### 3.2.1. Distributed Filtering

The data of each sensors are processed using KF [15,35] to omit the noise before being sent to the BS for inference, as shown in Figure 2a,b. KF is mainly based on the prediction operation which constructs the matrix of the underlying state vectors and updates it according to the sensor measurement. Consider the following model of the linear dynamic system. Table 1 explains all the notations used in the KF.
(2)$${\hat{x}}_{k}=A_{k}{\hat{x}}_{\left(k-1\right)}+B_{k}u_{k}+G_{k}$$
(3)$$z_{k}=H_{k}{\hat{x}}_{k}+v_{k}$$
(4)$${\hat{\dot{x}}}_{k}=A_{k}{\hat{x}}_{\left(k-1\right)}+Bu_{k}$$
(5)$${\hat{\dot{P}}}_{k}=A_{k}P_{k-1}A^{T}+Q$$

As shown in Equations (2)–(5), the estimation ${\hat{\dot{x}}}_{k}$ and covariance of error ${\hat{\dot{P}}}_{k}$ are integrated with the sensor measurement $z_{k}$ and covariance $R_{k}$ to get the updated estimate and error covariance matrix. Figure 2a,b explains the relation among the approximation and cleaned data, and Figure 3 shows the procedure of KF.
(6)$$K_{k}={\hat{\dot{P}}}_{k}H^{T}{\left(H{\hat{\dot{P}}}_{k}H^{T}+R\right)}^{-1}$$
(7)$${\hat{x}}_{k}={\hat{\dot{x}}}_{k}+K_{k}\left(z_{k}-H_{k}{\hat{\dot{x}}}_{k}\right)$$
(8)$$P_{k}=(1-K_{k}H){\hat{\dot{P}}}_{k}$$

#### 3.2.2. Bayesian Inference

The proposed scheme has two modes of operation: inference with dependent contexts and independent contexts, respectively. These two modes properly deal with heterogeneous data based on the relation between the contexts and data.

Bayesian Inference with Dependent Contexts

This mode is applied when the events or measurements of the environment are dependent on each other. Here, the probability of an event or outcome is predicted based on the occurrence of the related events. Making a decision based on multi-sensor data and context information is regarded as inference with the given condition. For example, turning an air conditioner on and off according to the existence of people or fire detection based on multiple context data are such inference operations. Similarly, in the healthcare application, a variety of sensor nodes record the patient’s physical context such as blood pressure and sugar, etc. [10]. When the threshold of a context is exceeded, the medical center is notified immediately. In critical applications, including the healthcare system, accurate and reliable inference is very important, and single-sensor data may not be sufficient for that [36,37]. Therefore, BI using heterogeneous data is required. In this mode, the posterior probability is computed according to the Bayesian rules to make the decision based on the compound contexts. Table 2 lists the notation used in the proposed CCBI model.

Figure 4 shows the data flow of the proposed CCBI with the change in the state of the system. Here, $z_{t}=(z_{t}^{1},z_{t}^{2},\;\ldots ,z_{t}^{n})$ is the set of data generated by n sensors, and $c_{t}=(c_{t}^{1},c_{t}^{2},\;\ldots ,c_{t}^{k})$ is a set of k context data at state t. The probability distribution, $p\left(z_{t}|c_{t}\right),$ represents how the contextual information affects the sensor reading and final event, $a_{t}.$ The state takes a certain probability value, which is expressed by the state transition model, $p\left(a_{t}|z_{t},c_{t},y_{t}\right),$ and leads to a final decision. The belief of the BI system, $Bel\left(a_{t}\right),$ under a specific state, $a_{t},$ and contextual information, $c_{t},$ is defined as
(9)$$Bel\left(a_{t}\right)=p\left(a_{t}|z_{n},c_{t},y_{t}\right),$$

The equation of Bayes fusion [38] is adopted for deriving the belief. Particularly, Equation (9) can be expressed as follows. Let us define $p\left(a_{t}\right)$ as the probability of the occurrence of an event, $a_{t},$ and $p\left(a_{t},z_{t},c_{t},y_{t}\right)$ as the joint probability of the occurrence of all events. Then, the conditional probability of the occurrence of $a_{t}$ given that the environment or State_$y_{t}$ has already occurred can be related as $p\left(a_{t},z_{t},c_{t},y_{t}\right)=p\left(a_{t}|z_{t},c_{t},y_{t}\right)\;p\left(z_{t},c_{t},y_{t}\right)$
(10)$$p\left(a_{t}|z_{t},c_{k},y_{t}\right)=\frac{p\left(a_{t},z_{t},c_{t},y_{t}\right)\;}{p\left(z_{t},c_{t},y_{t}\right)}$$

The following belief represents the conditional dependence between $a_{t}$ and $\left(z_{t},c_{t},y_{t}\right),$ and the above relation can also be written as follows in Equation (11):(11)$$p\left(a_{t}|z_{t},c_{t},y_{t}\right)=\frac{p\left(z_{t}|c_{t},y_{t},a_{t}\right)\;p\left(c_{t},y_{t},a_{t}\right)\;}{p\left(z_{t},c_{t},y_{t}\right)}$$

By applying the chain rule, $p\left(c_{t},y_{t},a_{t}\right)$ and $p\left(z_{t},c_{t},y_{t}\right)$ can be further separated as $p\left(c_{t},y_{t},a_{t}\right)=p\left(c_{t}|y_{t},a_{t}\right)p\left(y_{t},a_{t}\right),\;\mathrm{and}\;p\left(y_{t},a_{t}\right)=p\left(y_{t}|a_{t}\right)p\left(a_{t}\right)$, then
(12)$$p\left(c_{t},y_{t},a_{t}\right)=p\left(c_{t}|y_{t},a_{t}\right)p\left(y_{t}|a_{t}\right)p\left(a_{t}\right)$$

In a similar fashion, $p\left(z_{t},c_{t},y_{t}\right)$ can be formulated as $p\left(z_{t},c_{t},y_{t}\right)=p\left(z_{t}|c_{t},y_{t}\right)p\left(c_{t},y_{t}\right)\;\mathrm{and}\;p\left(c_{t},y_{t}\right)=p\left(c_{t}|y_{t}\right)p\left(y_{t}\right)$, then
(13)$$p\left(z_{t},c_{t},y_{t}\right)=p\left(z_{t}|c_{t},y_{t}\right)p\left(c_{t}|y_{t}\right)p\left(y_{t}\right)$$

Substituting Equations (12) and (13) into Equation (11), we obtain the following equation:(14)$$p\left(a_{t}|z_{n},c_{k},y_{t}\right)=\frac{p\left(z_{n}|c_{t},y_{t},a_{t}\right)\;p\left(c_{t}|y_{t},a_{t}\right)p\left(y_{t}|a_{t}\right)p\left(a_{t}\right)}{p\left(z_{t}|c_{t},y_{t}\right)p\left(c_{t}|y_{t}\right)p\left(y_{t}\right)}$$

By substituting Equation (14), Equation (9) becomes
(15)$$Bel\left(a_{t}\right)=\frac{p\left(z_{n}|c_{t},y_{t},a_{t}\right)\;p\left(c_{t}|y_{t},a_{t}\right)p\left(y_{t}|a_{t}\right)p\left(a_{t}\right)}{p\left(z_{t}|c_{t},y_{t}\right)p\left(c_{t}|y_{t}\right)p\left(y_{t}\right)}$$

The data service module produces the meta-information on the system to extract the information relevant to the target application such as updating the alarm threshold or reconfiguring the sensory infrastructure.

Bayesian Inference with Independent Contexts

This case is applied when the events or measurements of the sensor are not dependent on each other. After performing the KF operation, individual sensors send data to the central node, where the data are inferred based on the posterior and prior probability, and the measurement of the highest probability is considered as the final value. Let us denote $d_{n}^{i},$ $d_{c}^{i},$ and $d_{p}^{i}$ as the new predicted dataset, current dataset, and previous dataset with sensor-i, respectively $\left(i=1,2,3\ldots n\right)$. Note that the sensor data are individually filtered by KF, and the probability of the result, x, is computed based on the latest set of data $p\left(x|d_{n}^{1}\ldots d_{n}^{2}\right)\;$in the node applying the CCBI. Using the rules of CCBI, the following relationship can be obtained with two sensor nodes as in [39]:(16)$$Bel\left(x_{j}\right)=p\left(x|d_{n}^{1}d_{n}^{2}\right)=p\left(x|d_{c}^{1}d_{c}^{2}d_{p}^{1}d_{p}^{2}\right)=\frac{p\left(d_{c}^{1}d_{c}^{2}|x,d_{p}^{1}d_{p}^{2}\right)p\left(x|d_{p}^{1}d_{p}^{2}\right)}{p\left(d_{c}^{1}d_{c}^{2}|d_{p}^{1}d_{p}^{2}\right)}$$

Since the sensor readings are independent, we obtain the following:(17)$$Bel\left(x_{j}\right)=\frac{p\left(d_{c}^{1}|x,d_{p}^{1}\right)p\left(d_{c}^{2}|x,d_{p}^{2}\right)p\left(x|d_{p}^{1}d_{p}^{2}\right)}{p\left(d_{c}^{1}d_{c}^{2}|d_{p}^{1}d_{p}^{2}\right)}$$

By applying the chain rule, Equation (16) is rewritten as
$$Bel\left(x_{j}\right)=\frac{p\left(d_{c}^{1}|x,d_{p}^{1}\right)p\left(d_{c}^{2}|x,d_{p}^{2}\right)p\left(x|d_{p}^{1}d_{p}^{2}\right)}{p\left(d_{c}^{1}d_{c}^{2}|d_{p}^{1}d_{p}^{2}\right)}$$
(18)$$\;\;\;=\frac{p\left(x|d_{n}^{1}\right)p\left(x|d_{n}^{2}\right)p\left(x|d_{p}^{1}d_{p}^{2}\right)}{p\left(x|d_{p}^{1}\right)p\left(x|d_{p}^{2}\right)}$$

Equation (13) can be expanded for the case of three sensors as given below:(19)$$p\left(x|d_{n}^{1}d_{n}^{2}d_{n}^{3}\right)\;\;=\frac{p\left(x|d_{n}^{1}\right)p\left(x|d_{n}^{2}\right)p\left(x|d_{n}^{3}\right)p\left(x|d_{p}^{1}d_{p}^{2}d_{p}^{3})\right)}{p\left(x|d_{p}^{1}\right)p\left(x|d_{p}^{2}\right)p\left(x|d_{p}^{3}\right)}\times N_{3}$$
$$N_{3}=\overset{3}{\underset{i=1}{\sum }}p\left(x^{i}\right)$$

Here, $N_{3}$ denotes the normalization value for three sensor nodes. With m sensors, the model can be generalized as
(20)$$p\left(x|d_{n}^{1},\ldots ,d_{n}^{m}\right)=\frac{\prod_{i=1}^{m}p\left(x|d_{n}^{m}\right)p(x|\prod_{i=1}^{m}d_{p}^{m})}{\prod_{i=1}^{m}p\left(x|d_{p}^{m}\right)}\times N_{n}$$
(21)$$N_{m}=\overset{m}{\underset{i=1}{\sum }}p\left(x^{i}\right)$$

## 4. Performance Evaluation

In this section, we present the simulation environment as well as performance analysis of the obtained results.

### 4.1. Simulation Environment

The simulation environment is set up on a computer with Intel-Core i7 processor and 16 GB RAM running Matlab R2018a at LINK lab, koreatech university, South Korea. The performance of the CCBI is compared with the two representative BI schemes: Bayesian Data Fusion (BDF) [22] and Bayesian Network Data Fusion (BNDF) [23].

Bayesian Data Fusion (BDF): This is the simplest model of data fusion based on Bayes’ rule combining different knowledge. $P\left(x|z\right)$ is the inference distribution of the unknown state x using specific sensor measurement z. It is represented as
(22)$$P\left(x|z\right)=\alpha P\left(x\right)P(z|x)$$

The BDF model is further extended by BNDF to improve the inference accuracy as follows.

Bayesian Network Data Fusion (BNDF): This merges some properties of the surrounding environment, e, with the data for achieving more accurate inference as below [23]:(23)$$P\left(x|z,e\right)=\alpha P\left(x\right)P(z|x,e)$$

Note that the BNDF scheme is extended in the proposed CCBI scheme by adding different factors and states of the environment to improve the inference accuracy, as explained in the previous section.

The fire detection system is adopted to evaluate the efficiency of the BI schemes as a realistic test case. In fire detection, the key contexts are air temperature, smoke concentration, the wavelength of the flame radiation, humidity, and carbon monoxide (CO). The data used in the simulation are generated randomly for each of the five different contexts. In the simulation, gaussian noisy data of high variation are injected, and then erroneous data are omitted by applying KF [40]. The filtered data from different sensors are then sent to the CCBI engine to make a final decision based on the context information. Four performance indicators are measured in the simulation: sensitivity (precision), specificity (recall), accuracy, and F1-score [41]. The results of three schemes are computed and tested using three statistical measurements, sensitivity, specificity, and F1-score (F-measure), as shown in Equations (24)–(27). In the following equations, we used TP (true positive), FP (false positive), TN (true negative), and FN (false negative).
(24)$$Sensitivity\;\left(\mathrm{precision}\right)\;\;\;=\frac{TP}{TP+FN}$$
(25)$$specificity\;\left(\mathrm{recall}\right)=\frac{TN}{FP+TN}$$
(26)$$Accuracy=\frac{TP+TN\;}{TP+FP+TN+FN}$$
(27)$$F1=\frac{2\left(specificity\times sensitivity\right)}{\left(specificity+sensitivity\right)}$$

### 4.2. Simulation Result

Figure 5 shows the probability of fire detection with each respective context datum. As observed from the figure, at iteration 18, the detection probability with air temperature, smoke, flame, CO, and humidity is around 0.66, 0.61, 0.42, 0.44, and 0.4, respectively. When the fire erupts, the values of the contexts will start to increase, such as at iteration 10. However, the fire eruption can be confirmed between iterations 40 and 70 because the probability values of all the contexts are high enough in that range. Notice that making a decision based on only one context might not allow dependable results. For example, high temperatures without smoke may not indicate fire in the real situation. Therefore, the proposed scheme utilizes all the five-context data to efficiently resolve this issue and increase the decision accuracy.

Figure 6 shows the evaluation of the error rate in the sensor data before and after the KF. In the sensor data, errors occur due to noises and variations. To test the effectiveness of the proposed scheme, noisy data of high variation are injected in the simulation. Before the BI process, the sensor data, which are a mixture of the data and noise, are filtered by KF. It measures the noise variance in the data to filter out the errors, as briefly explained in Equations (4)–(8) in Section 3.2. As shown in the figure, the error rate after KF is much smaller than before KF. In addition, note that the error rate fluctuates significantly, while KF makes it quite stable and low.

Figure 7 shows the ROC (receiver operating characteristic) curve for three schemes, BDF, BNDF, and CCBI. Here, the curve is plotted with TP rate (=$TP/\left(TP+FN\right)$) on the y-axis against the FP rate (=$FP/\left(FP+TN\right)$) on the x-axis; it compares the TP and FP rate of three schemes. The upper-left corner indicates a 100% TP rate. Therefore, the closer to this corner, the higher the overall accuracy is. Hence, the proposed scheme shows better performance than the other two schemes, BDF and BNDF.

Figure 8 shows four performance measurements: the recall, precision, accuracy, and F1-score of the three schemes. The CCBI consistently shows better results than BDF and BNDF based on four performance measurements: recall, precision, accuracy, and F1-score. This better performance is because the decision-making strategy of the proposed scheme is based on multiple contexts.

## 5. Conclusions

In this paper, a novel, context-aware scheme of multi-sensor data inference was proposed for a smart IoT environment. It employed KF and BI to efficiently deal with the uncertainty and inconsistency issues with sensory data. To enable early warnings of fire, the characteristics of temperature, smoke concentration, and carbon monoxide sensor data in the initial stage of fire were analyzed in this study, and a BI was chosen to achieve the fusion of data. Here, the data were inferred considering the relation between the contexts. The simulation results show that the proposed scheme considerably outperformed the existing BI schemes in terms of decision accuracy, with a realistic test case of fire detection.

In the future, the performance of the proposed method will be further extended and improved by employing the Dempster–Shafer theory, which calculates the reliability of each state based on data extracted from various sources. The proposed scheme will also be extended using feature extraction and selection in the high-dimensional data to improve inference and decision accuracy. Moreover, it is also our goal to build a wireless sensor network (WSN) based on Deep Reinforcement Learning to test the algorithm in a realistic fire scenario and apply it to the Internet of Things (IoT) in smart cities.

## Figures and Tables

**Figure 1 sensors-22-03022-f001:**
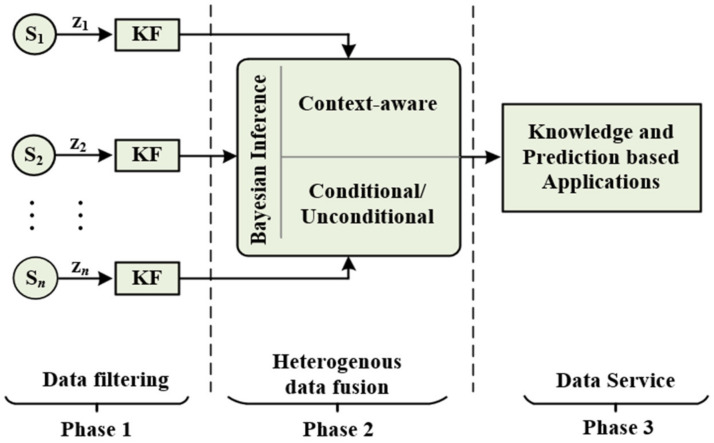
The two-phase operation of the proposed CCBI scheme.

**Figure 2 sensors-22-03022-f002:**
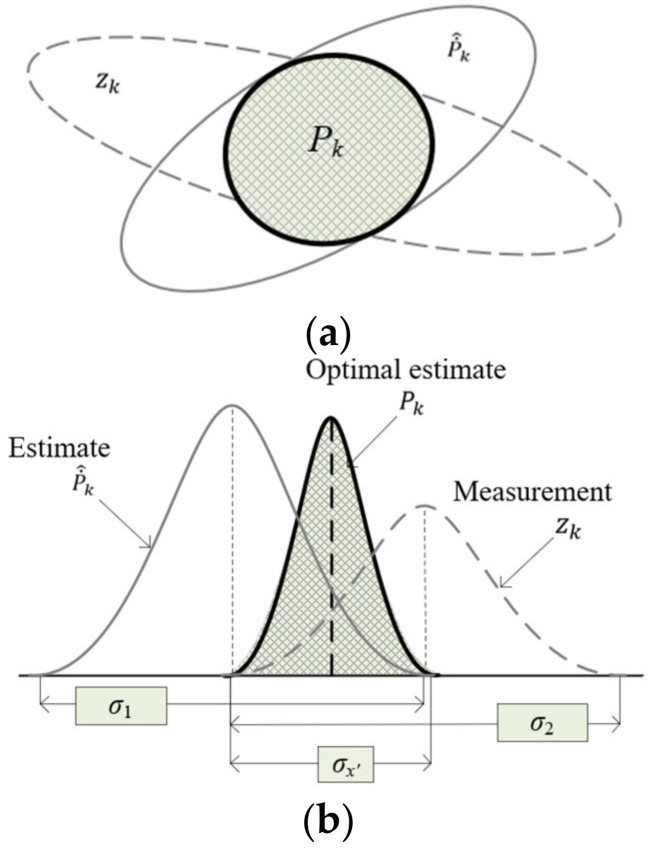
(**a**) The intersection of the covariance of data. (**b**) The gaussian PDFs of the estimation and measurement.

**Figure 3 sensors-22-03022-f003:**
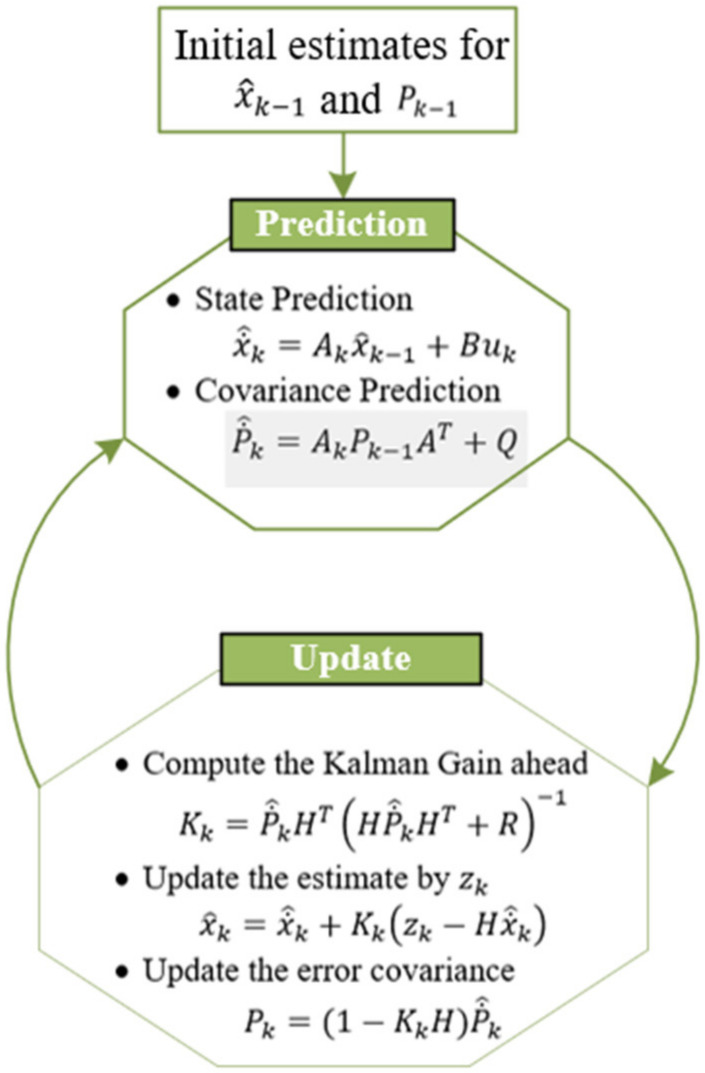
The operation procedure of KF.

**Figure 4 sensors-22-03022-f004:**
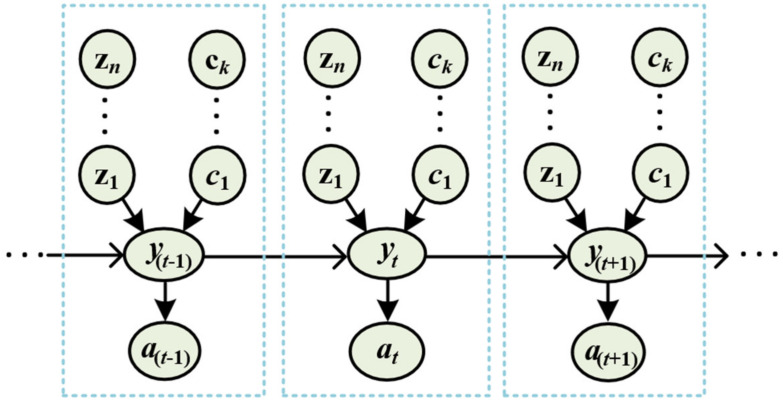
The data flow of the proposed CCBI.

**Figure 5 sensors-22-03022-f005:**
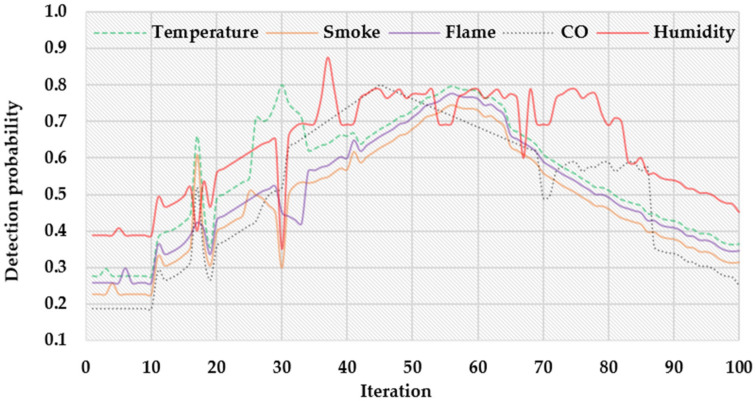
The detection probability of fire is based on the respective context.

**Figure 6 sensors-22-03022-f006:**
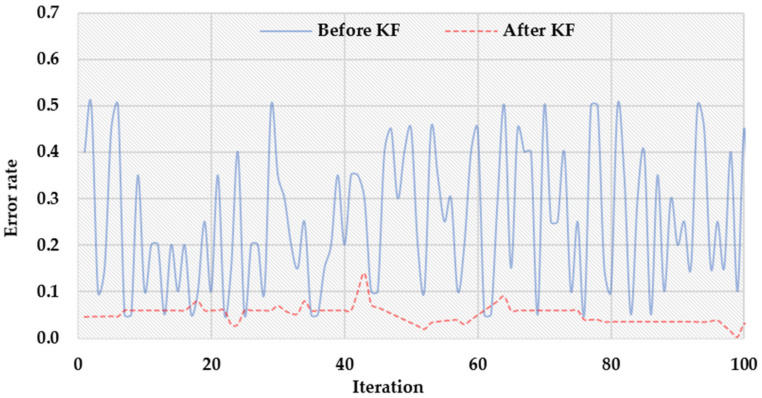
The comparison of error rates before and after KF.

**Figure 7 sensors-22-03022-f007:**
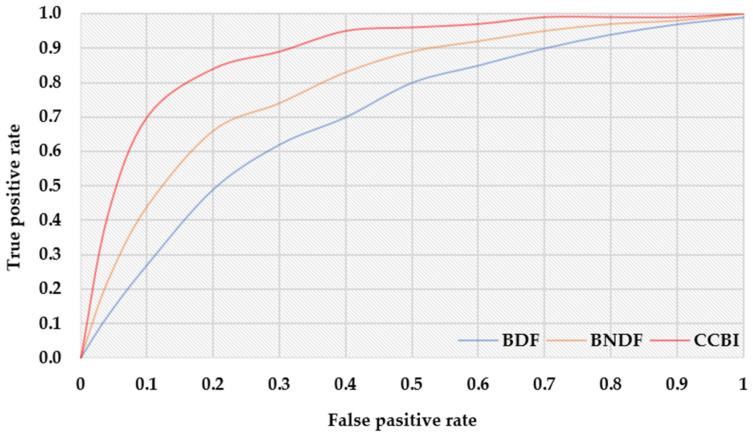
The result of TP and FP for three schemes.

**Figure 8 sensors-22-03022-f008:**
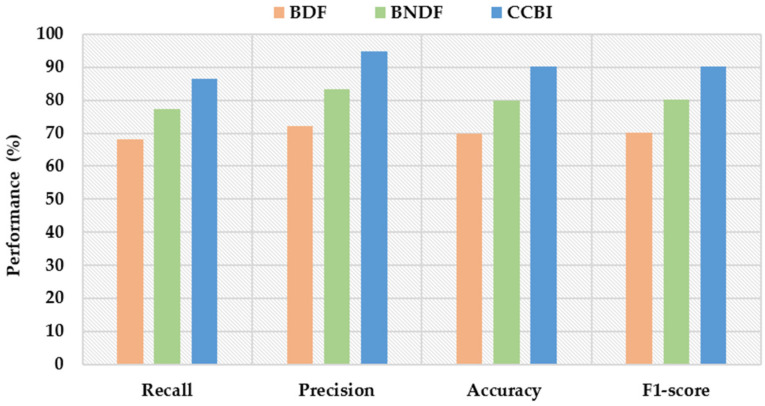
The comparison of the schemes on the four metrics.

**Table 1 sensors-22-03022-t001:** The notations used in KF.

${\hat{x}}_{k}$	The state of the process
$A_{k}$	The system state matrix
$B_{k}$	The Input matrix
$u_{k}$	The control vector
$G_{k}$	The process noise or gain
$z_{k}$	The measurement obtained by sensors
$H_{k}$	The Observation (model) matrix
$v_{k}$	Noise measurement or error
${\hat{\dot{x}}}_{k}$	The estimation of the predicted state
${\hat{\dot{P}}}_{k}$	Covariance of error
$R_{k}$	Covariance
$K_{k}$	The Kalman gain

**Table 2 sensors-22-03022-t002:** The notations used in the CCBI model.

$z_{t}$	Denotes the sensory measurement at state t
$c_{t}$	Represents the contextual information at state t
$y_{t}$	Denotes the environment at state t
$a_{t}$	Represents the target alarm value at state t
$p\left(.\right)$	Denotes the probability function on the measurement
$Bel\left(.\right)$	Represents the belief of the occurrence

## Data Availability

Not applicable.

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
