# Peer review of "Compound Context-Aware Bayesian Inference Scheme for Smart IoT Environment"

_sensors, 2022, doi:10.3390/s22083022_

Round 1
Reviewer 1 Report
The authors propose a context-aware scheme based on Bayesian inference to achieve accurate fusion and inference for Internet of Things application environments. In the proposed scheme, sensor data are fused based on the relation and contexts of data whether they are dependent or not on each other. The topic is up to date and relevant for the journal. My main concerns about the article can be found next.
The Figure 1 is too basic and does not add much to the technical side of the article. I suggest the authors to remove the Figure 1 and present the IoT features by text.
I miss a discussion about the Context Sharing paradigm. Context Sharing represents the scenarios in which context information needs to be shared among two or more instances for a reasoning process. As the proposed solution works with more than one context, it should mention the Context Sharing paradigm.
The authors compare Bayesian Data Fusion (BDF) and Bayesian Network Data Fusion (BNDF) results. Those are classic methods with pretty dated references. I suggest the authors compare their results with newer solutions (even extensions of the already compared solutions) or compare recent work that uses either BDF or BNDF in IoT application environments.
Author Response
The response to the reviewers comments are attached. Please see the attachment.

Reviewer 2 Report
Overall the paper seems to be interesting and sound, however, in my opinion, it is still affected by some problems.
First of all, in some parts, the clarity and editorial quality of the paper weaken. As a consequence, such parts result to be quite difficult to read. Therefore, I would suggest to carefully improve the prose of writing in order to make this paper easier to read.
Furthermore, presentation aside, by reading the paper, it still was not entirely clear what to expect with the direction of the article. Indeed, the contribution proposed in this paper has been only marginally compared and contextualised with respect to the state of the art. As a result, it is extremely difficult to understand the novelty/contributions introduced by the paper. The aforementioned aspects should be carefully addressed before the paper can be considered any further.
The paper should be better compared and contextualized with respect to the state of the art. I want suggest these papers to authors:
- Lombardi, M.; Pascale, F.; Santaniello, D. Internet of Things: A General Overview between Architectures, Protocols and Applications. Information 2021, 12, 87. https://doi.org/10.3390/info12020087
- Clarizia F., Colace F., Lombardi M., Pascale F., Santaniello D. (2018) A Multilevel Graph Approach for Road Accidents Data Interpretation. In: Castiglione A., Pop F., Ficco M., Palmieri F. (eds) Cyberspace Safety and Security. CSS 2018. Lecture Notes in Computer Science, vol 11161. Springer, Cham. https://doi.org/10.1007/978-3-030-01689-0_24
The paper results to be quite difficult to follow, due to the massive (and sometimes not adequately described) use of mathematical notation. The use of mathematical notation should always be supported by an appropriate informal description. By doing this, the paper may be easier to read and follow.
The figures should be better explained in their component parts
Finally, a thorough proofreading would be suggested, since in the paper there are some typos and formatting issues.
As remarks:
- The paper should be better compared and contextualized with respect to the state of the art.
- In some parts of the paper, the clarity and editorial quality of the paper weaken. As a consequence, such parts result to be quite difficult to read. Therefore, I would suggest to carefully improve the prose of writing in order to make this paper easier to read.
- Each figure should be properly defined within the text and must be improved in quality.
- Equations should be numbered appropriately and referenced within the text.
- An accurate proofreading is strongly recommended.
Round 2
Reviewer 2 Report
all suggestions have been applied, I recommend a reread to eliminate any typos
Author Response
Response to Reviewer 2 Comments (Round 2):
We highly regret for grammatical errors and typos. As the reviewer recommended, we have carefully read the whole manuscript for correcting the typos and grammatical errors. All the changes are highlighted is yellow. Please see the attached file.
Thank you very much for helpful review and comments.
